# Factors related to quality of life in community-dwelling adults in Sleman Regency, Special Region of Yogyakarta, Indonesia: Results from a cross-sectional study

**Fitrina Mahardani Kusumaningrum**[1,2☯]*, **Fatwa Sari Tetra Dewi**[1,3☯], **Ailiana Santosa**[4☯], **Heny Suseani Pangastuti**[5‡], **Polly Yeung**[6‡]

1 Faculty of Medicine, Department of Health Behavior, Public Health and Nursing, Environment and Social Medicine, Universitas Gadjah Mada, Yogyakarta, Indonesia, 2 Faculty of Medicine, Public Health and Nursing, Doctoral Program in Medicine and Health Science, Universitas Gadjah Mada, Yogyakarta, Indonesia, 3 Faculty of Medicine, Public Health and Nursing, Sleman Health and Demographic Surveillance System, Universitas Gadjah Mada, Yogyakarta, Indonesia, 4 Institute of Medicine, Sahlgrenska Academy, School of Public Health and Community Medicine, University of Gothenburg, Gothenburg, Sweden, 5 Faculty of Medicine, Medical-Surgical Nursing Department, Public Health and Nursing, Universitas Gadjah Mada, Yogyakarta, Indonesia, 6 School of Social Work, Massey University, Palmerston North, New Zealand

☯ These authors contributed equally to this work.
‡ These authors also contributed equally to this work
* fitrina.mahardani.k@mail.ugm.ac.id

**Data Availability Statement:** The data file in .xls format is available from the figshare database (DOI: 10.6084/m9.figshare.22152251).

## Abstract

### Background

Quality of life studies in low- and middle-income countries have demonstrated the influence of socioeconomic factors on the quality of life (QoL). However, further studies are required to confirm this association in developing countries with rapidly ageing populations. Using Ferrans et al.'s QoL model, this study aimed to identify the factors associated with the QoL of community-dwelling adults in Indonesia.

### Methods

A cross-sectional study among 546 community-dwelling adults aged 50+ years was conducted in Yogyakarta, Indonesia, in 2018. QoL was measured using the Short Form 12 questionnaire, which consists of a summary of physical and mental health. We performed stepwise logistic regression analyses to determine odds ratios (ORs) with 95% confidence intervals (CIs) and examined the association between the QoL (physical and mental health) and demographic characteristics, socioeconomic status, financial management behaviour, multimorbidity status, nutritional status, cognitive impairment status, depression status, and independence. Statistical significance was set at $p<0.05$.

### Results

Among the respondents, 15% reported poor physical health, and 9.2% reported poor mental health. Good physical health was significantly associated with the absence of chronic

**Funding:** The author (FSTD) received funding from Direktorat Riset dan Pengabdian Masyarakat, Kementerian Riset, Teknologi dan Pendidikan Tinggi, Republic of Indonesia (https://www.kemdikbud.go.id/) for this work, with the grant number: 1774/UN1/DITLIT/DIT-LIT/LT/2018. The funder had no role in study design, data collection and analysis, decision to publish, or preparation of the manuscript.

**Competing interests:** The authors have declared that no competing interests exist.

disease (OR 2.39; 95% CI: 1.07–5.33), independence in activities of daily living (OR 3.90; 95% CI 1.57–9.67) and instrumental activities of daily living (OR 4.34; 95% CI 2.28–8.26). Absence of depression was significantly associated with good mental health (OR 2.80; 95% CI 1.3–5.96).

## Conclusion

The QoL of community-dwelling adults in Indonesia is associated with activities of daily living and instrumental activities of daily living, as well as the absence of chronic disease and depression. Efforts should be made to prevent chronic disease and delay functional decline through healthy lifestyles and routine physical and mental health screenings.

## Introduction

The promotion of healthy ageing, defined as 'the process of developing and maintaining the functional ability that enables well-being in old age', is crucial to respond to the growing number of ageing populations [1]. Quality of life (QoL) is an outcome of healthy ageing and is described as a multidimensional concept covering health, mental, functional, emotional, and social aspects of life [2,3]. Identifying the determinants of QoL is important for supporting the development of healthy ageing interventions and providing appropriate care for older adults.

By 2050, approximately two-thirds of the global older adult population will reside in low- and middle-income countries (LMICs), with the highest increases observed in East and Southeast Asia [4]. Evidence on QoL and its determinants in LMICs has shown that sociodemographic, physical, psychological and social factors influence older adults' QoL [5–7]. However, a study conducted in two developing countries in Asia [8] suggested that socioeconomic factors may have a significant influence on QoL. Huang et al. corroborated this finding by identifying that financial stress may influence the QoL of older adults in developing countries [9]. However, further studies are required to explore the relationship between socioeconomic factors, especially financial factors, and QoL in developing countries with rapidly ageing populations and social challenges.

As one of the LMICs with the largest population in Southeast Asia [10], Indonesia is experiencing rapid demographic transition [11]. In 2022, the proportion of older adults was 10.48%, and is projected to reach 19.8% by 2045 [12,13]. In 2100, the old-age dependency ratio will reach approximately 50%, owing to the increase in older adult population and decreasing growth of the younger population [11]. In 2018, Sudharsanan and Bloom showed an upward trend in functional disability in a more recent cohort of older adults in Indonesia [14]. In addition to other issues, such as the increasing prevalence of chronic diseases across all ages, shrinking family sizes [10] and limited access to mental health care [15], older adults in Indonesia may experience considerable challenges in maintaining their QoL. Previous studies in Indonesia [16,17] and Southeast Asia [18,19] have identified that sociodemographic, physical, and psychological factors may influence QoL among older adults; however, these studies did not explore the influence of these determinants on QoL aspects. Exploration of the association between the determinants and QoL aspects is important to describe the dynamics of QoL and its related factors, and thus may provide recommendations for various disciplines.

In 2005, Ferrans et al. developed a model describing the relationship between QoL and individual and environmental characteristics, biological function, symptoms, functional status, and general health perceptions [20]. Ferrans et al. [20] modified the health-related QoL model

by Wilson and Cleary [21] by adding the link between individual and environmental characteristics to each concept that influences QoL, thus providing a more comprehensive description of the relationship between QoL and the associated factors. Given that the Ferrans et al. [20] model provides further understanding of the dynamics of QoL, this study utilized it and hypothesized that individual and environmental characteristics (sociodemographic factors, financial management behaviours, and living arrangements), symptoms (multimorbidity, depression, cognitive impairment, and nutritional status), and functional status (activities of daily living [ADLs] and instrumental activities of daily living [IADLs]) are related to the physical and mental health aspects of QoL.

QoL in older adults living in LMICs is associated with socioeconomic, social, psychological, and physical factors. However, the relationship between these factors and QoL aspects has rarely been explored [6,22]. In addition, there is a need for further exploration of the relationship between financial aspects and QoL in developing countries with rapidly ageing population [9]. Therefore, this study aimed to investigate the relationship between the physical and mental aspects of QoL and their determinants among community-dwelling adults in Indonesia. We included a sample of adults aged 50+ years to capture the dynamics of those nearing the retirement age in Indonesia (55 years in 2018) and their influence on QoL, as suggested in a previous study [11].

## Materials and methods

### Study design and setting

This community-based cross-sectional study was conducted between May and September 2018 in Sleman Regency, Special Region of Yogyakarta, Indonesia, which is located on Java, the most populous island in Indonesia. The province had the highest proportion of older adults (15.5%) in 2021 [23]. Sleman Regency is an administrative region in Yogyakarta, located on the slope of Merapi Mountain, one of the active volcanoes in Indonesia. In 2020, approximately 14% of individuals in Sleman Regency were older adults aged 60+ years [24]. Data from the Central Bureau of Statistics of Sleman Regency for 2022 showed that approximately 50% of older adults were still working, and approximately 35% of households were at risk of poverty. To cope with the ageing population, Sleman Regency has developed various community-based programs aimed at monitoring and promoting the health of older adults [25].

### Sampling and participants

The inclusion criteria for this study were adults 1) aged 50 years and older; 2) living in Sleman Regency; and 3) listed as respondents of the Sleman Health and Demographic Surveillance System (Sleman HDSS); the exclusion criteria were adults who declined to participate or were unavailable for two scheduled visits. The sample was randomly selected from a sample frame of the Sleman HDSS, a surveillance system initiated by the Faculty of Medicine, Public Health, and Nursing (FM-PHN), Universitas Gadjah Mada, Yogyakarta, Indonesia, which regularly collects demographic and health data from Sleman Regency. The Sleman HDSS collected data from 5147 households selected randomly from the census blocks of Sleman Regency using a two-stage cluster sampling method [26]. In 2018, 3129 older adults in Sleman Regency were registered in the Sleman HDSS sampling frame. The Sleman HDSS sampling frame was used to facilitate monitoring and follow-up in future studies.

To calculate the minimum sample size in this study, we applied the sample size formula for comparing proportions [27] using the proportions of male (89.4%) and female (80%) older adults with good QoL from a prior study [17], resulting in a total of 488 individuals with 80%

power. We added 75 individuals (15% of the total sample) in case of nonresponses and failed interviews, resulting in a minimal sample size of 563 adults.

## Variables

The outcome variable in this study was QoL, which is presented as physical and mental health. The exposure variables included demographic characteristics, socioeconomic status, financial management behaviour, nutritional status, multimorbidity status, cognitive impairment status, depression status, and independence (activities of daily living [ADLs] and instrumental activities of daily living [IADLs]), as suggested in previous studies [16,17,28].

*Quality of life* was assessed using the Short Form Health Survey (SF-12v2)–4 weeks recall questionnaire consisting of 12 questions measuring eight domains: physical functioning, role-physical, bodily pain, general health, vitality, social functioning, role-emotional, and mental health [29]. The score was developed using norm-based methods and standardized to the US population by developing indicator variables (coded 0 and 1) for each response, where a score of 1 indicated a favourable response. These indicator variables were then weighted and aggregated using the regression coefficient and intercept from the 1998 US population survey to calculate the physical and mental health scores. The total score from each summary was then transformed into norm-based scoring so that the score ranged from 0 to 100, with a standard deviation of 10 and a mean of 50. Using the PRO CoRE scoring software developed by OptumInsight Life Science, Inc., the physical and mental health scores were dichotomized into poor (score $\leq$ 50) and good (score > 50) health. Individuals with poor physical and mental health were used as the reference groups.

*Demographic characteristics* consisted of age (51–65, 66–80, $\geq$81 years), sex (male and female), marital status (married, or single/divorced/widowed), type of occupation (government/private sector, farmer/labourer, retired/other, or not working), education level (high, medium, low), and living arrangement (living with family member(s) or living alone).

*Socioeconomic status* was derived from information on household food and beverage consumption, household expenditure, homeownership, and home characteristics from the Sleman HDSS data. This information was used to calculate wealth index using principal component analysis. Higher scores indicate better socioeconomic status [26]. Socioeconomic status was classified into three groups (upper, middle, and lower).

*Financial management behaviour* was measured using information on savings and investments, insurance behaviour, cash management, and credit management. Responses ranged from 'never' (coded 1) to 'always' (coded 5). Respondents were given the option of 'not applicable', which was recorded as the lowest score ('never'). The responses were summarized and reverse coded [30] for items that reflected poor financial management behaviour. We calculated the scores using exploratory factor analysis. The median score was used to determine two categories of behaviour: good (above the median) and poor (below and within the median).

*Nutritional status* was assessed on the basis of nutritional intake and physical status. Physical conditions were determined using body mass index, upper arm circumference, and calf circumference, whereas nutritional intake was determined using questions about the number of meals and daily protein, fruits/vegetables, and fluids intake. The measurements were conducted three times to anticipate variances during the measurements, such as surveyor conditions (skills and exhaustion) and respondent postures, and the average score was determined. Each response was scored (range: 0–1, 0–2, and 0–3) with a higher score indicating a more favourable response. The total score was categorized as good (score > 24) or moderate to poor (score $\leq$ 24) [31].

*Multimorbidity* is defined by the World Health Organization as 'the coexistence of two or more chronic conditions in the same individual' [32]. We assessed the multimorbidity status

based on questions regarding the presence of chronic diseases (hypertension, diabetes mellitus, stroke, coronary heart disease, asthma, and chronic obstructive pulmonary disease) diagnosed by a health professional (yes or no) in the Sleman HDSS data. The total number of chronic diseases was calculated and categorized into three groups (no chronic diseases, one chronic disease, and two or more chronic diseases).

*Cognitive impairment status* was assessed using 30 questions covering orientation, repetition, verbal recall, attention and calculation, language, and visual construction (scores ranging from 0 to 30). We classified the scores into normal mild (score > 20) and moderate severe (score ≤ 20) groups using the standard cut-off point [33].

*Depression status* was assessed using 15 questions regarding respondents' feelings over the past two weeks [34]. Responses were summarized and classified into two categories: not at risk (score ≤ 4) and at risk or depressed (score ≥ 5). For respondents with severe cognitive impairment, a different questionnaire to assess their depression status [35] was distributed to those who had frequent contact with respondents. Responses were summarized and classified into two categories: not at risk (score ≤ 9) and at risk or depressed (score > 9).

*Independence* was assessed using six ADLs questions (bathing, dressing, eating, getting up from lying down, using the toilet, and getting to/going to the toilet) and four IADLs questions (taking care of household responsibilities, participating in community activities, using public or private transport, and going where one wants to go). Difficulty in performing the tasks was self-assessed using a Likert scale (none, mild, moderate, hard, and extreme difficulty). Those with no or mild difficulty in performing all tasks were categorized as independent, and those with moderate, hard, or extreme difficulty in performing at least one task were categorized as dependent.

## Instruments

We collected the data using a set of standardized instruments consisting of the following: SF-12v2 to assess QoL; the Financial Management Behaviour Scale to measure financial management behaviour; the Mini Nutritional Assessment to assess nutritional status; the Mini Mental State Examination to assess cognitive impairment status; the Geriatric Depression Scale to assess depression status among respondents without cognitive impairment [34]; the Cornell Scale of Depression for Dementia to assess depression status among respondents with severe cognitive impairment [35]; and part of World Health Organization Study on Global Ageing and Adult Health to assess the independence in ADLs and IADLs [36]. Before data collection, we conducted a pretest to assess the validity and reliability of the instrument in the study population. The analysis showed satisfactory results for each instrument, with Cronbach's α value ranging from 0.6 to 0.8.

## Data collection

During data collection, enumerators recruited adults using addresses from the Sleman HDSS database (described in more detail elsewhere) [26]. A face-to-face computer-assisted personal interview lasting approximately 67 minutes was conducted, in which trained enumerators asked the respondents to answer closed-ended questions and recorded their responses on a digital tablet. Relatives were allowed to accompany the respondents during the interviews; however, only the responses from the respondents were recorded. The quality of the data collection process was ensured through spot checks and cross checks by three data collection supervisors.

The Medical and Health Research Ethics Committee of the Faculty of Medicine, Public Health, and Nursing, Universitas Gadjah Mada, Yogyakarta, Indonesia, approved the project

(KE/FK/0501/EC/2018) in May 2018. Before data collection, the enumerators explained the study's objectives, procedures, risks, benefits, respondents' roles and rights, and data confidentiality to the respondents. The information was also written in a form that respondents were given time to read and sign if they agreed to participate. Each respondent provided written informed consent.

## Statistical analysis

Analyses were conducted using STATA 13.1 software (STATA Corp., College Station, TX, USA). Categorical variables were presented as counts and percentages, and continuous variables were presented as means and standard deviations. Univariate and multivariate logistic regression analyses were used to examine the association between determinants and physical and mental health scores. Five models were built using stepwise logistic regression: *Model 1* (adjusted for demographic factors), *Model 2* (Model 1 adjusted for socioeconomic status and financial management behaviour), *Model 3* (Model 2 adjusted for multimorbidity status, nutritional status, and cognitive impairment status), *Model 4* (Model 3 adjusted for depression status), and *Model 5* (Model 4 adjusted for independence, i.e. ADLs and IADLs). An interaction test was performed for sex and ADLs. The results are presented as odds ratios (ORs) with 95% confidence intervals (CIs) and $p<0.05$ to indicate significance. We conducted a multicollinearity test that showed minimum intercorrelation between the explanatory variables (variance inflation factor < 5 and tolerance factor > 0.2) [37].

## Results

Of the 563 respondents at the first visit, 546 were available to participate in the second visit. The majority of respondents were aged 51–65 years (52.4%), female (56%), married (65.6%), not working/retired (54.8%), had a low education level (63%), and lived with family member (s) (94.5%). Approximately 26% of respondents reported having at least one chronic disease, 12% were at risk of depression or feeling depressed, and 34% had moderate/severe cognitive impairment. Most respondents had good physical (84.3%) and mental (90.8%) health (Table 1); however, more respondents scored lower (below 50) on physical health than mental health (Fig 1).

### Physical health

The results of the univariate analysis showed that good physical health was significantly associated with all types of occupations, maintaining good financial management behaviour, good nutritional status, having no or one chronic disease, not being at risk of depression, having normal to mild cognitive impairment, and independence in ADLs and IADLs (Table 2).

After adjustment for covariates, good physical health was associated with not having chronic diseases (OR 2.39; 95% CI: 1.07–5.33) and being independent in both ADLs (OR 3.90; 95% CI: 1.57–9.67) and IADLs (OR 4.34; 95% CI: 2.28–8.26). After exploring the interaction between sex and ADLs, women who were independent in ADLs had 6.89 higher odds (95% CI: 1.82–26.04) of experiencing good physical health than dependent women. Stepwise regression analysis showed an improvement in the physical health model, with a final pseudo-$R^2$ value of 0.21 (Table 3).

### Mental health

As shown in Table 4, in the unadjusted regression model, good mental health was associated with being aged 51–65 years, male, working as a farmers or labourer, being retired or other,

**Table 1. Demographic characteristics, socioeconomic status, financial management behaviour, nutritional status, multimorbidity status, cognitive impairment status, depression status, independence, and quality of life of the respondents.**

| Variables | Group | n | % |
|---|---|---|---|
| **Demographic** | | | |
| Age in Years | 51–65 | 286 | 52.4 |
| | 66–80 | 219 | 40.1 |
| | 81+ | 41 | 7.5 |
| Sex | Male | 240 | 44.0 |
| | Female | 306 | 56.0 |
| Marital Status | Married | 358 | 65.6 |
| | Single/divorced/widowed | 188 | 34.4 |
| Occupation | Government/private sector | 82 | 15.0 |
| | Farmer/labourer | 165 | 30.2 |
| | Retired/other | 81 | 14.8 |
| | Not working | 218 | 40.0 |
| Education Level | High education level | 37 | 6.8 |
| | Medium education level | 165 | 30.2 |
| | Low education level | 344 | 63.0 |
| Living Arrangement | Living with family member(s) | 516 | 94.5 |
| | Living alone | 30 | 5.5 |
| **Socioeconomic Status** | Upper | 120 | 22.0 |
| | Middle | 176 | 32.2 |
| | Lower | 250 | 45.8 |
| **Financial Management Behaviour** | Good | 273 | 50.0 |
| | Poor | 273 | 50.0 |
| **Nutritional Status** | Good | 398 | 72.9 |
| | Moderate–poor | 148 | 27.1 |
| **Multimorbidity** | No chronic diseases | 350 | 64.1 |
| | 1 chronic disease | 145 | 26.6 |
| | 2 or more chronic diseases | 51 | 9.3 |
| **Cognitive Impairment Status** | Normal–mild | 362 | 66.3 |
| | Moderate–severe | 184 | 33.7 |
| **Depression Status** | Not at risk | 480 | 87.9 |
| | At risk or depressed | 66 | 12.1 |
| **Independence in ADLs** | Independent | 497 | 91.0 |
| | Dependent | 49 | 9.0 |
| **Independence in IADLs** | Independent | 365 | 66.9 |
| | Dependent | 181 | 33.1 |
| **Quality of Life** | | | |
| Physical health | Good | 460 | 84.3 |
| | Poor | 86 | 15.8 |
| Mental health | Good | 496 | 90.8 |
| | Poor | 50 | 9.2 |

ADLs, activities of daily living; IADLs, instrumental activities of daily living.

having a medium level of education, having an upper socioeconomic status, having good financial management behaviours and nutritional status, having no history of depression and having normal to mild cognitive impairment. Additionally, good mental health was associated with independence in ADLs and IADLs (Table 4).

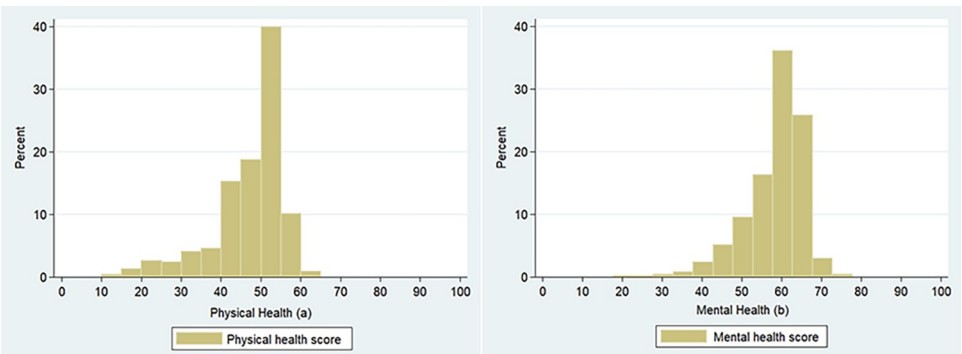

**Fig 1. Frequency distribution of physical and mental health scores.** (a) physical health score. (b) mental health score.

After controlling for other variables, good mental health was only associated with not being at risk for depression (OR 2.80; 95% CI: 1.32–5.96). Stepwise regression analysis showed improvement in the mental health models with a final pseudo-$R^2$ value of 0.15 (Table 5).

## Discussion

Using various determinants, this study explored QoL, both in terms of physical and mental health, in one of Southeast Asia's LMICs where data are limited. Overall, we found that respondents had higher mean scores for mental health than for physical health. The absence of chronic disease and independence in ADLs and IADLs were associated with good physical health, while the absence of depression was associated with good mental health. Women who were independent in ADLs had better physical health than those who were dependent.

In contrast to previous studies in developing countries, we did not find any association between socioeconomic status or financial management behaviour and the physical or mental health aspects of QoL. Multivariate analysis showed an association between financial management behaviour and physical health; however, this association did not persist after adding physical symptom variables, such as multimorbidity, cognitive impairment status, and nutritional status. We also found an association between occupation type and physical health, but the association did not persist after adjusting for functional status. Previous studies have shown that the influence of socioeconomic status on QoL is mediated by social functioning, social support, and depressive symptoms [38,39]. Our limited sample may not have been able to identify this association. Previous studies also suggested that the impact of multimorbidity and functional limitations in Indonesia is expanding [14,40], which may explain the greater influence of physical symptoms and functional limitations compared to socioeconomic variables.

Higher mean mental health scores than physical health scores are commonly found in studies on older adults in community settings [41,42]. Older adults may experience better mental health because mental health scores improve during middle age while physical health continues to decline [43,44]. As age increases, individuals may master more competencies and strategies to cope with emotional experiences by adjusting their goals and gaining wisdom and resilience. In the Indonesian context, this wisdom includes individuals believing that God will provide the solution to their problems if the individuals keep making their best effort ("*semeleh*" and "*ikhtiar*") [45,46]. In general, older adults tend to direct their attention towards more emotionally meaningful goals, resulting in better emotional regulation to maintain good mental health [47].

**Table 2. Univariate analysis of explanatory variables of good physical health.**

| Variables | | Physical Health | | | | Unadjusted OR (95% CI) | p value |
|---|---|---|---|---|---|---|---|
| | | Good | | Poor | | | |
| | | n | % | n | % | | |
| **Age in Years** | 51–65 | 249 | 54.1 | 37 | 43.0 | 1.26 (0.56–2.84) | 0.580 |
| | 66–80 | 179 | 38.9 | 40 | 46.5 | 1.89 (0.84–4.28) | 0.125 |
| | 81+ # | 32 | 7.0 | 9 | 10.5 | | |
| **Sex** | Male | 203 | 44.1 | 37 | 43.0 | 1.05 (0.66–1.67) | 0.849 |
| | Female# | 257 | 55.9 | 49 | 57.0 | | |
| **Marital Status** | Married | 308 | 67.0 | 50 | 58.1 | 1.46 (0.91–2.34) | 0.116 |
| | Single/divorced/widowed# | 152 | 33.0 | 36 | 41.9 | | |
| **Occupation** | Government/private sector | 76 | 16.5 | 6 | 7.0 | 4.27 (1.76–10.4) | **0.001**** |
| | Farmer/labourer | 150 | 32.6 | 15 | 17.4 | 3.37 (1.83–6.23) | **0.000***** |
| | Retired/other | 71 | 15.4 | 10 | 11.6 | 2.40 (1.16–4.97) | **0.019*** |
| | Not working# | 163 | 35.4 | 55 | 64.0 | | |
| **Education Level** | High education level | 32 | 7.0 | 5 | 5.8 | 1.30 (0.50–3.47) | 0.603 |
| | Medium education level | 142 | 30.9 | 23 | 26.7 | 1.25 (0.74–2.11) | 0.400 |
| | Low education level# | 286 | 62.1 | 58 | 67.4 | | |
| **Living Arrangement** | With family member(s) | 436 | 94.8 | 80 | 93.0 | 1.36 (0.54–3.44) | 0.513 |
| | Alone# | 24 | 5.2 | 6 | 7.0 | | |
| **Socioeconomic Status** | Upper | 105 | 22.8 | 15 | 17.4 | 1.58 (0.84–2.96) | 0.155 |
| | Middle | 151 | 32.8 | 25 | 29.1 | 1.36 (0.80–2.32) | 0.801 |
| | Lower# | 204 | 44.4 | 46 | 53.5 | | |
| **Financial Management** | Good | 243 | 52.8 | 30 | 34.9 | 2.09 (1.29–3.38) | **0.003**** |
| | Poor# | 217 | 47.2 | 56 | 65.1 | | |
| **Nutritional Status** | Good | 351 | 76.3 | 47 | 54.7 | 2.67 (1.66–4.30) | **0.000***** |
| | Moderate–poor# | 109 | 23.7 | 39 | 45.3 | | |
| **Multimorbidity** | No chronic diseases | 310 | 67.4 | 40 | 46.5 | 4.23 (2.18–8.20) | **0.000***** |
| | 1 chronic disease | 117 | 25.4 | 28 | 32.6 | 2.28 (1.12–4.62) | **0.022*** |
| | 2 or more chronic diseases# | 33 | 7.2 | 18 | 20.9 | | |
| **Depression Status** | Not at risk | 412 | 89.6 | 68 | 79.1 | 2.27 (1.25–4.14) | **0.007**** |
| | At risk or depressed# | 48 | 10.4 | 18 | 20.9 | | |
| **Cognitive Impairment Status** | Normal–mild | 315 | 68.5 | 47 | 54.7 | 1.80 (1.13–2.88) | **0.014*** |
| | Moderate–severe# | 145 | 31.5 | 39 | 45.3 | | |
| **ADLs** | Independent | 435 | 94.6 | 62 | 72.1 | 6.74 (3.62–12.5) | **0.000***** |
| | Dependent# | 25 | 5.4 | 24 | 27.9 | | |
| **IADLs** | Independent | 337 | 73.3 | 28 | 32.6 | 5.68 (3.46–9.32) | **0.000***** |
| | Dependent# | 123 | 26.7 | 58 | 74.4 | | |

* Significant at $p < 0.05$

** significant at $p < 0.01$

*** significant at $p < 0.001$; OR, odds ratio; CI, confidence interval

# reference group; ADLs, activities of daily living; IADLs, instrumental activities of daily living.

Our results showed that the absence of depression was associated with good mental health. This finding is consistent with those of previous studies [48] that explored the association between depression and QoL. Depression in older adults tends to be overlooked because of its unfamiliar symptoms and similarity with grief for the loss of relatives or important kin. In Indonesia, depression among older adults is associated with poor physical function and self-

**Table 3. Multivariate analysis of explanatory variables of good physical health.**

| Variables | Model I OR (95% CI) | Model II OR (95% CI) | Model III OR (95% CI) | Model IV OR (95% CI) | Model V OR (95% CI) |
|---|---|---|---|---|---|
| **Age in Years** | | | | | |
| 51–65 | 1.00 (0.41–2.44) | 1.06 (0.43–2.61) | 0.98 (0.37–2.55) | 1.05 (0.40–2.76) | 0.71 (0.25–2.03) |
| 66–80 | 0.96 (0.41–2.22) | 1.06 (0.45–2.49) | 1.08 (0.44–2.64) | 1.16 (0.47–2.87) | 0.96 (0.37–2.48) |
| 81+[#] | | | | | |
| **Sex** | | | | | |
| Male | 0.59 (0.34–1.03) | 0.69 (0.39–1.22) | 0.63 (0.35–1.14) | 0.63 (0.35–1.14) | 0.31 (0.07–1.43) |
| Female[#] | | | | | |
| **Marital Status** | | | | | |
| Married | 1.21 (0.69–2.11) | 1.20 (0.68–2.12) | 1.14 (0.63–2.05) | 1.18 (0.66–2.13) | 1.03 (0.55–1.93) |
| Single/divorced/ widowed[#] | | | | | |
| **Occupation** | | | | | |
| Government/private sector | 4.80 (1.88–12.25)* | 3.95 (1.52–10.27)* | 3.48 (1.31–9.25)* | 3.42 (1.28–9.12)* | 2.44 (0.86–6.96) |
| Farmer/labourer | 3.91 (1.99–7.68)* | 4.03 (2.05–7.90)* | 3.28 (1.63–6.60)* | 3.22 (1.60–6.48)* | 2.04 (0.97–4.29) |
| Retired and other | 3.02 (1.22–7.44)* | 2.43 (0.97–6.12) | 2.21 (0.85–5.73) | 2.17 (0.83–5.67) | 2.02 (0.71–5.72) |
| **Not working[#]** | | | | | |
| **Educational Level** | | | | | |
| High education level | 0.78 (0.24–2.52) | 0.49 (0.14–1.76) | 0.40 (0.11–1.48) | 0.38 (0.10–1.41) | 0.47 (0.12–1.84) |
| Medium education level | 1.13 (0.63–2.04) | 0.89 (0.47–1.68) | 0.76 (0.38–1.53) | 0.75 (0.37–1.50) | 0.71 (0.34–1.48) |
| **Low education level[#]** | | | | | |
| **Living Arrangements** | | | | | |
| With family member(s) | 1.12 (0.41–3.07) | 1.00 (0.35–2.83) | 1.05 (0.36–3.05) | 0.95 (0.33–2.80) | 1.08 (0.35–3.27) |
| Alone[#] | | | | | |
| **Socioeconomic Status** | | | | | |
| Upper | | 1.36 (0.62–2.96) | 1.50 (0.67–3.39) | 1.55 (0.68–3.55) | 1.47 (0.62–3.51) |
| Middle | | 1.29 (0.72–2.32) | 1.29 (0.70–2.35) | 1.33 (0.73–2.45) | 1.17 (0.61–2.24) |
| Lower[#] | | | | | |
| **Financial Management Behaviour** | | | | | |
| Good | | 1.93 (1.11–3.37)* | 1.57 (0.88–2.80) | 1.59 (0.89–2.83) | 1.38 (0.79–2.87) |
| Poor[#] | | | | | |
| **Nutritional Status** | | | | | |
| Good | | | 2.29 (1.27–4.13)* | 2.14 (1.18–3.90)* | 1.50 (0.79–2.86) |
| Moderate–poor[#] | | | | | |
| **Multimorbidity** | | | | | |
| No chronic diseases | | | 3.19 (1.55–6.57)* | 3.06 (1.48–6.36)* | 2.39 (1.07–5.33)* |
| 1 chronic disease | | | 1.60 (0.73–3.52) | 1.54 (0.70–3.39) | 1.16 (0.49–2.76) |
| 2 or more chronic diseases[#] | | | | | |
| **Cognitive Impairment Status** | | | | | |
| Normal–mild | | | 1.09 (0.59–2.02) | 1.05 (0.57–1.95) | 0.66 (0.33–1.31) |
| Moderate–severe[#] | | | | | |
| **Depression Status** | | | | | |
| Not at risk | | | | 1.70 (0.86–3.36) | 1.57 (0.75–3.28) |
| At risk or depressed[#] | | | | | |
| **Activities of Daily Living** | | | | | |
| Independent | | | | | 3.90 (1.57–9.67)* |
| Dependent[#] | | | | | |
| **Instrumental Activities of Daily Living** | | | | | |

*(Continued)*

**Table 3.** (Continued)

| Variables | Model I OR (95% CI) | Model II OR (95% CI) | Model III OR (95% CI) | Model IV OR (95% CI) | Model V OR (95% CI) |
|---|---|---|---|---|---|
| Independent | | | | | 4.34 (2.28–8.26)* |
| Dependent[#] | | | | | |
| **Interaction (Sex and Activities of Daily Living)** | | | | | |
| Male#independent | | | | | 1.76 (0.35–8.81) |
| Male#dependent | | | | | 0.54 (0.28–1.08) |
| Female#independent | | | | | 6.89 (1.82–26.0)* |
| Female#dependent[#] | | | | | |
| Pseudo $R^2$ | 0.0619 | 0.0774 | 0.1819 | 0.2027 | 0.2063 |

* Significant at $p < 0.05$; OR, odds ratio; CI, confidence interval; [#] reference group.

rated health [49]. Integrating mental health screening into the available health services for older adults may be important for improving QoL in community settings.

Interestingly, we did not find an association between mental health and the presence of chronic diseases or ADLs, although this study was conducted in a country with limited mental health services and professionals [15]. This finding contradicts that of a previous study conducted in China [6]. One possible explanation for this is that older adults may become more resilient, defined as their ability to cope with and adapt to stressful events, which may mask the effects of multimorbidity and difficulty in ADLs [50]. When coping with physical health problems, adults in Indonesia rely on religious and spiritual approaches, such as trying to understand the meaning of their conditions, submitting to God's will and eventually accepting their fate (*'pasrah'* or *'nrimo'*) [51]. These practices were conducted to gain comfort and lessen the effect of illness on their mental states, as shown in patients with diabetes mellitus in Surabaya, Indonesia [52]. Studies have shown that this practice can improve emotional well-being; however, for some individuals, it can also lead to fatalistic attitudes that prevent active involvement in disease management and worsen physical health [51,53]. Therefore, there is a need to promote active involvement in disease management while considering spiritual and religious approaches to maintaining good mental health.

Our results indicated that independence in performing ADLs and IADLs and the absence of chronic diseases were correlated with physical health. This result corroborates the findings of a previous study in Indonesia on QoL among adults with chronic diseases, such as diabetes mellitus [54], chronic obstructive pulmonary disease [55], and cardiovascular diseases [56]. In Indonesia, maintaining good physical health may also be challenging, as older adult populations experience an upward trend in functional disability in earlier cohorts, which may be caused by the increasing prevalence of multimorbidity [14]. When dealing with chronic diseases, Indonesian individuals also tend to delay accessing health care until they experience symptoms that affect their daily activities. They also prefer self-care and self-medication before consulting a professional [57], which may worsen their physical condition and result in functional disability.

For women, independence in ADLs is important to maintain good physical health. As women live longer than men, ADL dependence due to the progression of age may greatly affect women [58,59]. In Indonesia, women are more likely to experience difficulties in bathing, going to the toilet, and dressing than men [60], which may decrease physical health to a greater extent than other types of ADLs [41]. As women are generally responsible for preparing food and doing other household chores, losing independence in ADLs may impact older women's

**Table 4. Univariate analysis of explanatory variables of good mental health.**

| Variables | | Mental Health | | | | Unadjusted OR (95% CI) | p value |
|---|---|---|---|---|---|---|---|
| | | Good | | Poor | | | |
| | | n | % | n | % | | |
| **Age in Years** | 51–65 | 266 | 53.6 | 20 | 40.0 | 2.74 (1.08–6.95) | **0.034*** |
| | 66–80 | 196 | 39.5 | 23 | 46.0 | 1.75 (0.70–4.41) | 0.232 |
| | 81+ # | 34 | 6.9 | 7 | 14.0 | | |
| **Sex** | Male | 225 | 45.4 | 15 | 30.0 | 1.94 (1.03–3.64) | **0.040*** |
| | Female# | 271 | 54.6 | 35 | 70.0 | | |
| **Marital Status** | Married | 331 | 66.7 | 27 | 54.0 | 1.71 (0.95–3.07) | 0.073 |
| | Single/divorced/widowed# | 165 | 33.3 | 23 | 46.0 | | |
| **Occupation** | Government/private sector | 76 | 15.3 | 6 | 12.0 | 2.10 (0.84–5.24) | 0.112 |
| | Farmer/labourer | 154 | 31.1 | 11 | 22.0 | 2.32 (1.13–4.76) | **0.022*** |
| | Retired/other | 79 | 15.4 | 2 | 4.0 | 6.55 (1.53–28.0) | **0.011*** |
| | Not working# | 187 | 35.4 | 31 | 62.0 | | |
| **Education Level** | High education level | 36 | 7.3 | 1 | 2.0 | 5.01 (0.67–37.5) | 0.117 |
| | Medium education level | 158 | 31.8 | 7 | 14.0 | 3.14 (1.38–7.15) | **0.006**** |
| | Low education level# | 302 | 60.9 | 42 | 84.0 | | |
| **Living Arrangement** | With family member(s) | 471 | 95.0 | 45 | 90.0 | 2.09 (0.76–5.73) | 0.151 |
| | Alone# | 25 | 5.0 | 5 | 10.0 | | |
| **Socioeconomic Status** | Upper | 117 | 23.6 | 3 | 6.0 | 5.12 (1.53–17.2) | **0.008**** |
| | Middle | 158 | 31.8 | 18 | 36.0 | 1.15 (0.62–2.15) | 0.656 |
| | Lower# | 221 | 44.6 | 29 | 58.0 | | |
| **Financial Management** | Good | 257 | 51.8 | 16 | 32.0 | 2.29 (1.23–4.25) | **0.009**** |
| | Poor# | 239 | 48.2 | 34 | 68.0 | | |
| **Nutritional Status** | Good | 374 | 75.4 | 24 | 48.0 | 3.32 (1.84–6.00) | **0.000***** |
| | Moderate–poor# | 122 | 24.6 | 26 | 52.0 | | |
| **Multimorbidity** | No chronic diseases | 323 | 65.1 | 27 | 54.0 | 1.90 (0.78–4.63) | 0.156 |
| | 1 chronic disease | 129 | 26.0 | 16 | 32.0 | 1.28 (0.50–3.32) | 0.608 |
| | 2 or more chronic diseases# | 44 | 8.9 | 7 | 14.0 | | |
| **Depression Status** | Not at risk | 445 | 89.7 | 35 | 70.0 | 3.74 (1.91–7.31) | **0.000***** |
| | At risk or depressed# | 51 | 10.3 | 15 | 30.0 | | |
| **Cognitive Impairment Status** | Normal–mild | 342 | 68.9 | 20 | 40.0 | 3.33 (1.83–6.05) | **0.000***** |
| | Moderate–severe# | 154 | 31.1 | 30 | 60.0 | | |
| **ADLs** | Independent | 458 | 92.3 | 39 | 78.0 | 3.40 (1.61–7.17) | **0.001**** |
| | Dependent# | 38 | 7.7 | 11 | 22.0 | | |
| **IADLs** | Independent | 340 | 68.6 | 25 | 50.0 | 2.18 (1.21–3.92) | **0.009**** |
| | Dependent# | 156 | 31.4 | 25 | 50.0 | | |

* Significant at $p<0.05$

** significant at $p<0.01$

*** significant at $p<0.001$; OR, odds ratio; CI, confidence interval

# reference group; ADLs, activities of daily living; IADLs, instrumental activities of daily living.

ability to take care of their families and themselves, which may also affect their overall well-being [60].

Our results highlight the importance of comprehensive and holistic programs covering the physical and mental health of older adults. Programs that prevent chronic diseases and delay functional decline by promoting healthy lifestyles and routine screening for physical and mental health may improve the QoL of older adults in Indonesia. The current program that can

**Table 5. Multivariate analysis of explanatory variables of good mental health.**

| Variables | Model I OR (95% CI) | Model II OR (95% CI) | Model III OR (95% CI) | Model IV OR (95% CI) | Model VI OR (95% CI) |
|---|---|---|---|---|---|
| **Age in Years** | | | | | |
| 51–65 | 1.56 (0.56–4.34) | 1.60 (0.57–4.48) | 1.16 (0.39–3.48) | 1.42 (0.46–4.34) | 1.52 (0.48–4.77) |
| 66–80 | 1.35 (0.52–3.49) | 1.51 (0.57–3.95) | 1.29 (0.47–3.51) | 1.53 (0.55–4.26) | 1.61 (0.57–4.55) |
| 81+[#] | | | | | |
| **Sex** | | | | | |
| Male | 1.24 (0.62–2.50) | 1.47 (0.71–3.02) | 1.30 (0.62–2.73) | 1.34 (0.63–2.84) | 1.34 (0.63–2.86) |
| **Female**[#] | | | | | |
| **Marital Status** | | | | | |
| Married | 1.00 (0.50–1.98) | 0.94 (0.47–1.90) | 0.87 (0.43–1.78) | 0.94 (0.46–1.94) | 0.94 (0.45–1.97) |
| Single/divorced/ widowed[#] | | | | | |
| **Occupation** | | | | | |
| Government/private sector | 1.51 (0.58–3.97) | 1.22 (0.45–3.32) | 1.06 (0.38–2.93) | 0.97 (0.34–2.74) | 1.03 (0.35–2.97) |
| Farmer/labourer | 2.11 (0.96–4.64) | 2.22 (1.00–4.88)* | 2.03 (0.90–4.59) | 1.94 (0.85–4.41) | 2.05 (0.86–4.85) |
| Retired and other | 3.00 (0.59–15.19) | 2.29 (0.44–11.98) | 2.05 (0.38–11.13) | 1.82 (0.33–10.03) | 1.90 (0.35–10.39) |
| Not working[#] | | | | | |
| **Educational Level** | | | | | |
| High education level | 2.84 (0.32–25.12) | 1.11 (0.11–11.39) | 0.88 (0.08–9.45) | 0.76 (0.07–8.30) | 0.67 (0.06–7.43) |
| Medium education level | 2.47 (1.02–5.96)* | 1.85 (0.74–4.63) | 1.37 (0.52–3.62) | 1.32 (0.33–10.03) | 1.29 (0.48–3.43) |
| Low education level[#] | | | | | |
| **Living Arrangements** | | | | | |
| With family member(s) | 1.52 (0.51–4.53) | 1.52 (0.49–4.71) | 1.49 (0.47–4.72) | 1.24 (0.38–4.02) | 1.45 (0.44–4.80) |
| Alone[#] | | | | | |
| **Socioeconomic Status** | | | | | |
| Upper | | 3.10 (0.80–11.95) | 3.10 (0.79–12.15) | 3.46 (0.85–13.97) | 3.60 (0.87–14.86) |
| Middle | | 0.93 (0.47–1.84) | 0.88 (0.44–1.78) | 0.96 (0.47–1.94) | 0.93 (0.45–1.91) |
| Lower[#] | | | | | |

(*Continued*)

**Table 5.** (Continued)

| Variables | Model I OR (95% CI) | Model II OR (95% CI) | Model III OR (95% CI) | Model IV OR (95% CI) | Model VI OR (95% CI) |
|---|---|---|---|---|---|
| **Financial Management Behaviour** | | | | | |
| Good | | 1.82 (0.91–3.62) | 1.47 (0.72–2.99) | 1.57 (0.77–3.21) | 1.66 (0.80–3.44) |
| Poor[#] | | | | | |
| **Nutritional Status** | | | | | |
| Good | | | 2.05 (1.03–4.10)* | 1.79 (0.88–3.65) | 1.80 (0.86–3.75) |
| Moderate–poor[#] | | | | | |
| **Multimorbidity** | | | | | |
| No chronic diseases | | | 1.54 (0.59–4.06) | 1.31 (0.48–3.59) | 1.22 (0.43–3.49) |
| 1 chronic disease | | | 0.97 (0.34–2.78) | 0.82 (0.27–2.45) | 0.78 (0.25–2.43) |
| 2 or more chronic diseases[#] | | | | | |
| **Cognitive Impairment Status** | | | | | |
| Normal–mild | | | 1.76 (0.86–3.63) | 1.64 (0.79–3.40) | 1.71 (0.79–3.63) |
| Moderate–severe[#] | | | | | |
| **Depression Status** | | | | | |
| Not at risk | | | | 2.86 (1.36–6.00)* | 2.80 (1.32–5.96)* |
| At risk or depressed[#] | | | | | |
| **Activities of Daily Living** | | | | | |
| Independent | | | | | 1.79 (0.73–4.37) |
| Dependent[#] | | | | | |
| **Instrumental Activities of Daily Living** | | | | | |
| Independent | | | | | 0.67 (0.30–1.52) |
| Dependent[#] | | | | | |
| **Pseudo R$^2$** | 0.066 | 0.088 | 0.118 | 0.139 | 0.146 |

* Significant at $p<0.05$; OR, odds ratio; CI, confidence interval

[#] reference group.

accommodate these needs is the Integrated Health Care Post for Older Adults. However, this program mainly focuses on monitoring the physical health of older adults and lacks support and participation [61,62]. Evidence from LMICs has shown that bottom-up interventions tailored to the population's needs and sensitive to local culture are promising and merit exploration [63], especially in Southeast Asia, where social, cultural, and economic conditions are diverse [64]. In Indonesia, because of the collectivist culture, support from the community or religious leaders is important to encourage the adoption of a healthy lifestyle and disease management in the community. A case study in Yogyakarta showed that a healthy ageing program could be successful with support from community leaders, shared beliefs about the importance of the program and shared beliefs about non-material rewards of volunteerism [25].

Our study provides the following suggestions. First, there is a clear need for a comprehensive and holistic health care program that covers both the physical and mental health of older adults. Second, healthy ageing programs in Indonesia should prevent chronic disease and delay functional decline by promoting the benefits of a healthy lifestyle and providing routine screening for both physical and mental health. Third, bottom-up interventions that acknowledge community needs and values are required, especially for Southeast Asian LMICs.

Our study's strength lies in the exploration of older adults' QoL in physical and mental health summary findings using comprehensive instruments to shape interventions that promote older adults' QoL in LMICs, especially in Southeast Asia, where data are limited. We also examined the association of each determinant with physical and mental health to better understand the dynamics of older adults' QoL. Although the data were collected in 2018, the variables collected in this study were considered to have slow changes in the population over the years [65,66]; thus, the data are still relevant for testing our hypothesis.

However, this study has some limitations. As we used a cross-sectional design, we could not demonstrate a causal relationship between QoL and independent variables. This study also lacked information about social support, social capital, and health behaviours that may further explain the physical and mental health of older populations. The use of the Sleman HDSS sampling frame and the limited sample size also limited our ability to observe associations between the variables that have been found in other studies. This study was conducted before the COVID-19 pandemic, which may affect the status and dynamics of QoL. Further studies with larger sample sizes are needed to explore these associations in greater detail. Future studies should also explore the dynamics of QoL in cultural and social contexts, preferably using qualitative methods, to inform the development of a healthy ageing program that is sensitive to the needs of older population and local values.

## Conclusion

Independence in ADLs and IADLs and the absence of chronic disease and depression are associated with older adults' QoL in Indonesia. Interventions and programs aimed at improving older adults' QoL should promote chronic disease prevention and management, delaying functional decline and preventing depression. These interventions and programs should be comprehensive, holistic, sensitive to cultural values and tailored to community needs.

## Acknowledgments

We would like to express our gratitude to the respondents who were willing to participate in this study, Ms. Putri Tiara Rosha who assisted in the data collection process, the Sleman HDSS professionals who provided aid and support during data collection and the Faculty of Medicine, Public Health and Nursing, Universitas Gadjah Mada.

## Author Contributions

**Conceptualization:** Fitrina Mahardani Kusumaningrum, Fatwa Sari Tetra Dewi, Heny Suseani Pangastuti, Polly Yeung.

**Data curation:** Fitrina Mahardani Kusumaningrum, Ailiana Santosa.

**Formal analysis:** Fitrina Mahardani Kusumaningrum, Ailiana Santosa.

**Funding acquisition:** Fatwa Sari Tetra Dewi.

**Investigation:** Fitrina Mahardani Kusumaningrum.

**Methodology:** Fitrina Mahardani Kusumaningrum, Fatwa Sari Tetra Dewi, Ailiana Santosa, Heny Suseani Pangastuti, Polly Yeung.

**Project administration:** Fitrina Mahardani Kusumaningrum, Fatwa Sari Tetra Dewi.

**Resources:** Fitrina Mahardani Kusumaningrum, Fatwa Sari Tetra Dewi.

**Supervision:** Fatwa Sari Tetra Dewi, Heny Suseani Pangastuti, Polly Yeung.

**Validation:** Fitrina Mahardani Kusumaningrum, Fatwa Sari Tetra Dewi, Ailiana Santosa.

**Visualization:** Fitrina Mahardani Kusumaningrum, Fatwa Sari Tetra Dewi, Ailiana Santosa.

**Writing – original draft:** Fitrina Mahardani Kusumaningrum, Fatwa Sari Tetra Dewi, Ailiana Santosa.

**Writing – review & editing:** Fitrina Mahardani Kusumaningrum, Fatwa Sari Tetra Dewi, Ailiana Santosa, Heny Suseani Pangastuti, Polly Yeung.

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
