## [Decision Letter · Decision Letter 0]

23 May 2023

PONE-D-23-05720Factors related to quality of life in community-dwelling older adults in Indonesia: Summary results from a cross-sectional studyPLOS ONE

Dear Dr. Kusumaningrum,

Thank you for submitting your manuscript to PLOS ONE. After careful consideration, we feel that it has merit but does not fully meet PLOS ONE’s publication criteria as it currently stands. Therefore, we invite you to submit a revised version of the manuscript that addresses the points raised during the review process.

Please consider all suggestions made by reviewers and have the revised manuscript reviewed by a native English speaker.

We look forward to receiving your revised manuscript.

Kind regards,

Fabíola Bof de Andrade

Academic Editor

PLOS ONE

Journal Requirements:

Additional Editor Comments:

In addition, I suggest the authors provide more information and review the topics below.

1 – Please, provide more information about the criteria used to classify the participants into good or poor physical and mental. Although the authors mentioned that they used a software it is needed to report the methods applied and how they defined the cut-off points.

2 – Line 244 and line 2060r. Please, inform what test is being used to define the amount of explanation by these variables.

3 – Since this is a cross-sectional study the authors should avoid using the word predict to refer to the associations between the variables.

4 – Line 279 – “In the Indonesian context, this wisdom includes individuals believing that God will provide the solution to their problems if the individual keeps trying with their best effort (“semeleh” and “ikhtiar”). Please, provide a reference to support the sentence.”

5 - Line 296 – “When coping with physical health problems, older adults in Indonesia rely on religious and spiritual approaches, such as trying to understand the meaning of their conditions, submitting to God’s will and eventually accepting their fate (‘pasrah’ or ‘nrimo’).” Please, provide a reference to support the sentence.

6 Line – 314 “As the person in the family who is responsible with preparing food and other household chores, losing independence in ADLs may impact older women’s ability to take care of their family and themselves, and this may also affect their overall QoL.” Please, provide a reference to support the sentence.

7 – I suggest the authors critically evaluate the findings regarding the variables that were not significant in the models such as the socioeconomic ones.

Reviewers' comments:

Reviewer's Responses to Questions

**Comments to the Author**

1. Is the manuscript technically sound, and do the data support the conclusions?

Reviewer #1: No

Reviewer #2: Partly

2. Has the statistical analysis been performed appropriately and rigorously? 

Reviewer #1: Yes

Reviewer #2: Yes

3. Have the authors made all data underlying the findings in their manuscript fully available?

Reviewer #1: No

Reviewer #2: No

4. Is the manuscript presented in an intelligible fashion and written in standard English?

Reviewer #1: No

Reviewer #2: Yes

5. Review Comments to the Author

Reviewer #1: Dear authors, the topic of the present manuscript is quite interesting; however, I have following comments that can be addressed:-

1. The introduction is not up to the mark; I think authors can try to write more comprehensive introduction along with a conceptual framework and well defined objectives and hypothesis.

2. More background about the geography of the study area can be defined.

3. How was the quality of the data controlled while face to face interview.

4. The tables can be well presented. The authors can refer to more research papers.

5. Did the authors used pre-regression tests like checking multi-collinearity.

6. I cannot see various models in the tables; kindly check.

7. The grammatical errors can rectified by getting the manuscript reviewed by professional english editor.

Reviewer #2: I am grateful for the opportunity to review the and could be contribute to enrich of the manuscript.

The theme is relevant and needs to be better studied in different contexts around the world.

I suggest some questions to bring greater clarity to the text and the proposed objectives.

My considerations follow attached.

Yours sincerely,

6. PLOS authors have the option to publish the peer review history of their article (what does this mean?). If published, this will include your full peer review and any attached files.

Reviewer #1: No

Reviewer #2: No

---

## [Author Response · Author response to Decision Letter 0]

25 Jul 2023

Response to Reviewers (PONE-D-23-05720)

Comments by the Editor

Methods

1. Please, provide more information about the criteria used to classify the participants into good or poor physical and mental. Although the authors mentioned that they used a software it is needed to report the methods applied and how they defined the cut-off points.

The authors thank the editor for highlighting the detail needed for this manuscript. We edited and added the following description in the manuscript to explain the process of developing the score of outcome variables: 

Line 149 – 155: “The score was developed using norm-based methods and standardized to the US population by developing indicator variables (coded 0 and 1) for each response, where a score of 1 indicated a favourable response. These indicator variables were then weighted and aggregated using the regression coefficient and intercept from the 1998 US population survey to calculate the physical health score and mental health score. The total score from each summary was then transformed into norm-based scoring, so that the score ranged from 0 to 100 and had a standard deviation of 10 and a mean of 50”.

Line 156 – 157: “physical and mental health scores were dichotomized into poor (score < 50) and good (score > 50) health”.

2. Line 244 and line 260. Please, inform what test is being used to define the amount of explanation by these variables.

Thank you for highlighting this unclear statement. The number of variables explanation was derived from the value of pseudo-R2 from the stepwise regression using the McFadden formula. However, we understand that the researcher should be cautious when using the value of pseudo-R2 to identify the amount of variable explanation because, unlike the R2 value derived from the ordinary least square approach, the pseudo-R2 value depend on the amount of sample and the number of variables. It was suggested that the pseudo-R2 value should only be interpreted to indicate model improvement. Therefore, we edited the statement as follows:

Line 268 – 269: “The stepwise regression analysis showed improvement of the physical health model with a final pseudo-R2 value of 0.21 (Table 3)”

Line 283 – 284: “The stepwise regression analysis showed improvement of the mental health models with a final pseudo-R2 value of 0.15 (Table 5)”

3. Since this is a cross-sectional study, the authors should avoid using the word predict to refer to the association between the variables.

We agree that we cannot derive a cause-and-effect relationship from a cross-sectional study. Therefore, we changed the word “predict” into “associated with” in our manuscript, which can be found in lines 291 – 292 and line 386.

Discussion

4. Line 279 – “In the Indonesian context, this wisdom includes individuals believing that God will provide the solution to their problems if the individual keeps trying with their best effort (“semeleh” and “ikhtiar”). Please, provide a reference to support the sentence.

Thank you for your suggestion. We added the reference to the statement in line 313.

5. Line 296 – “When coping with physical health problems, older adults in Indonesia rely on religious and spiritual approaches, such as trying to understand the meaning of their conditions, submitting to God’s will and eventually accepting their fate (‘pasrah’ or ‘nrimo’).” Please, provide a reference to support the sentence.

Thank you for your suggestion. We added the reference to the statement in line 330.

6. Line – 314 “As the person in the family who is responsible with preparing food and other household chores, losing independence in ADLs may impact older women’s ability to take care of their family and themselves, and this may also affect their overall QoL.” Please, provide a reference to support the sentence.

Thank you for your suggestion. We added the reference of the statement in line 349.

7. I suggest the authors critically evaluate the findings regarding the variables that were not significant in the models such as the socioeconomic ones.

Thank you for your kind suggestions. We added an explanation regarding the absence of association between socio-demographic variables and QoL in the discussion as follows:

Line 294 – 305: “In contrast to previous studies in developing countries, we did not find any association between socioeconomic status or financial management behaviour and physical or mental health aspects of QoL. The multivariate analysis showed an association between financial management behavior and physical health; however, this association did not persist after adding physical symptom variables, such as multimorbidity, cognitive impairment status and nutritional status. We also found an association between types of occupations and physical health, but it did not persist after adjustment for functional status. Previous studies identified that the influence of socioeconomic status on QoL was mediated by social functioning, social support, and depressive symptoms [35,36], and our limited sample may not have been able to identify this association. A previous study also suggested that the impact of multimorbidity and functional limitation in Indonesia is expanding [12,37], which may explain the greater influence of physical symptoms and functional limitation compared to socioeconomic variables”.

Comments by Reviewer#1

Introduction

1. The introduction is not up to the mark; I think authors can try to write more comprehensive introduction along with a conceptual framework and well-defined objectives and hypothesis.

Thank you for the reviews and kind suggestions. We acknowledge the unclear purpose and hypothesis of this study. Our literature review regarding quality of life (QoL) of older adults identified that, although the evidence is growing, studies in developing countries, especially in Southeast Asia are limited, as reported by Haraldstad et al. in 2019. In addition, the studies in Southeast Asia did not explore the relationship between the determinant and aspects or dimensions of QoL. However, this information is important to understand the dynamic of QoL and can provide recommendations for further studies and practices in specific disciplines, such as mental health practitioner or social scientist. 

We used the conceptual framework from Ferrans et al. (2005) which identified that QoL is influenced by the characteristics of environment and individual, general health perception, functional status, symptoms, and biological function. This framework is the modification of Wilson and Cleary’s (1995) model and warrants for further application . As the study in developing countries showed the great influence of socio-economic factors on QoL, applying this framework is appropriate because this framework identified the link between individual factors to QoL and other concepts influencing QoL. Therefore, we revised the introduction by adding information about the conceptual framework, aim and hypothesis of this study as follows:

Line 81 – 96: “This study applied the conceptual model developed by Ferrans et al. (2005), which describes the relationship between individual and environmental characteristics, biological function, symptoms, functional status, and general health perceptions and QoL. The model is a modification of Wilson and Cleary’s (1995) model that adds a link between individual and environmental characteristics to each concept that influences QoL [18]. Quality of life in older adults living in LMICs is associated with socioeconomic, social, psychological and physical factors. However, the relationship of these factors with QoL aspects has scarcely been explored [6,19]. In addition, there is a need for further exploration of the relationship between the financial aspect and QoL in developing countries that have a rapidly ageing population [9]. Therefore, this study aimed to investigate the relationship between the physical and mental aspects of QoL and its determinants among older community-dwelling adults in Indonesia. We hypothesized that individual and environmental characteristics (sociodemographic factors, financial management behaviours, and living arrangements), symptoms (multimorbidity, depression, cognitive impairment, and nutritional status) and functional status (activities of daily living (ADLs) and instrumental activities of daily living (IADLs)) are related to the physical and mental health aspects of QoL in older adults”. 

2. More background about the geography of the study area can be defined.

We added more description of the study area and the characteristics in the study design and setting section.

Line 101 – 109: “The Special Region of Yogyakarta is a province located on Java island, the most populous island in Indonesia. The province had the highest proportion of older adults (15.5%) in 2021 [20]. Sleman Regency is an administrative region in Yogyakarta, located on the slope of Merapi Mountain, one of the active volcanoes in Indonesia. In 2020, approximately 14% individuals in Sleman Regency were older adults aged 60+ [21]. Data from the Central Bureau of Statistics of Sleman Regency in 2022 showed that approximately 50% of older adults were still working and approximately 35% of households were at risk of poverty. In coping with the ageing population, Sleman Regency developed various community-based programs that aimed to monitor and promote older adults’ health [22]”.

3. Page 3, lines 69-70: What is the gap in the literature that other studies carried out in the South-Eastern Asian did not fill?

As mentioned earlier, we identified that there is limitation of studies in Southeast Asia and Indonesia which explored the relationship of determinants on each QoL aspect or dimensions. This information is essential to guide further studies for specific disciplines and understand the dynamic of QoL. Therefore, we edited and added more details in the following lines:

Line 75 – 80: “Previous studies in Indonesia [14,15] and Southeast Asia [16,17] identified that sociodemographic, physical and psychological factors may influence QoL among older adults; however, these studies did not explore the influence of these determinants on QoL aspects. Exploration of the association between the determinants and QoL aspects is important to describe the dynamics of QoL and its related factors, and thus may provide recommendations for various disciplines”.

4. Page 3, lines 79-80: Why does this study stand out in relation to other studies on quality of life in elderly people living in low- and middle-income countries? The innovation of the study regarding its contribution to the literature should be clearer. What is the relevance of the study to the context of public health?

Thank you for highlighting this aspect. From the literature review, we identified that socio-economic factors might have great influence on QoL in developing countries, as mentioned by Gosh et al in 2020. Huang et al., however, also identified the effect of financial stress on QoL among older adults in developing countries. Therefore, we added the variable of financial management behaviour, which from previous study was identified as the determinant of financial well-being, to complement the socio-economic status. This aspect is scarcely explored in the study of QoL determinants in developing countries. We added and edited our manuscript to cover this explanation:

Line 65 – 68: “Huang et al. (2020) corroborated this finding by identifying that financial stress may influence the QoL of older adults in developing countries. However, further studies are needed to explore the relationship between socioeconomic factors, especially financial aspect, and QoL in the developing countries that have rapidly ageing populations and social challenges [9]”.

Line 85 – 96: “Quality of life in older adults living in LMICs is associated with socioeconomic, social, psychological and physical factors. However, the relationship of these factors with QoL aspects has scarcely been explored [6,19]. In addition, there is a need for further exploration of the relationship between the financial aspect and QoL in developing countries that have a rapidly ageing population [9]. Therefore, this study aimed to investigate the relationship between the physical and mental aspects of QoL and its determinants among older community-dwelling adults in Indonesia. We hypothesized that individual and environmental characteristics (sociodemographic factors, financial management behaviours, and living arrangements), symptoms (multimorbidity, depression, cognitive impairment, and nutritional status) and functional status (activities of daily living (ADLs) and instrumental activities of daily living (IADLs)) are related to the physical and mental health aspects of QoL in older adults”.

Method

1. How was the quality of the data controlled while face to face interview?

We added information about maintaining data collection as follows:

Line 135 – 137: “The quality of the data collection process was ensured through spot checks and cross checks by three data collection supervisors”. 

Results

1. The tables can be well presented. The authors can refer to more research papers.

Thank you for your kind suggestion. We omitted Table 2 and edited the univariate table so that the table provide the p-value of the univariate analysis. We also added two tables describing the development of the model of physical and mental health variables.

2. Did the authors used pre-regression tests like checking multi-collinearity?

We conducted a multi-collinearity check and obtained the VIF and tolerance factor (1/VIF). We added the description in the manuscript as follows:

Line 240 – 241: “We conducted a multicollinearity test that showed minimum intercorrelation between explanatory variables (variance inflation factor (VIF) < 5 and tolerance factor > 0.2) [34]”.

3. I cannot see various models in the tables; kindly check.

We provide two additional table which describe the model development for both physical and mental health variables in line 270 and line 285, respectively.

Writings

1. The grammatical errors can rectified by getting the manuscript reviewed by professional english editor.

Thank you for your kind suggestion. The revised manuscript has been reviewed by a professional language editor.

Comments by Reviewer#2

Title

1.The title uses the term “summary results from a cross-sectional study”. This leads to the impression that the results are partial in relation to the study as a whole, and not that they are summarized only in relation to the quality of life outcome”. I suggest review.

Thank you for your kind suggestion. We understand the term “summary” may provide different impression regarding our study. Therefore, we omitted “summary” in our title to avoid misperception.

Abstract

1. In general, the summary brings important points such as the objectives, information about the study design, the population, the study period, the variables, the statistical method used, as well as the main results and conclusion of the study. However, I believe that the introduction of the abstract should be better explored. The context and justification of the study were not clear. Is the importance of studying quality of life among older adults living in low- and middle-income countries related only to the fact that research and data are limited in these contexts? This observation also applies to the introduction.

We edited the background of the abstract in the following lines:

Line 26 – 30: “Quality of life studies in low- and middle-income countries have shown the influence of socioeconomic factors on quality of life. However, there is a need for further studies to identify this association in a developing country with a rapidly ageing population. Using Ferrans and colleagues’ model of quality of life, this study aimed to identify the factors associated with the quality of life of community-dwelling older adults in Indonesia”.

Introduction

1. Page 3, line 67: The authors cite studies about factors associated with quality of life among older adults/elderly people in other contexts of low- and middle-income countries. It would be important to briefly mention the results of the cited studies, mainly to characterize the health-related quality of life in LMICs and to differentiate from high-income countries.

Thank you for your kind suggestions and further references to enrich our manuscript. We appreciate your consideration of our study. We edited our introduction to describe the results of former study in LMICs and identified some results which characterize the QoL of older adults in developing countries as follows:

Line 62 – 68: “Evidence on QoL and its determinants in LMICs identified that sociodemographic, physical, psychological and social factors influence older adults’ QoL [5–7]. A study in two developing countries in Asia [8], however, suggested that socioeconomic factors may have a great influence on QoL. Huang et al. (2020) corroborated this finding by identifying that financial stress may influence the QoL of older adults in developing countries. However, further studies are needed to explore the relationship between socioeconomic factors, especially financial aspect, and QoL in the developing countries that have rapidly ageing populations and social challenges [9]”.

2. Page 3, lines 69-70: What is the gap in the literature that other studies carried out in the South-Eastern Asian did not fill?

Studies in Southeast Asia identified that further studies in different populations are needed. In addition, the previous study in developing countries in the Southeast Asia region did not explore the relationship between the determinants and QoL aspects or dimensions. However. this information is important to provide recommendations for further practices and studies from various disciplines, such as mental health practitioners or social scientist. We added these descriptions in the following statement:

Line 75 – 80: “Previous studies in Indonesia [14,15] and Southeast Asia [16,17] identified that sociodemographic, physical and psychological factors may influence QoL among older adults; however, these studies did not explore the influence of these determinants on QoL aspects. Exploration of the association between the determinants and QoL aspects is important to describe the dynamics of QoL and its related factors, and thus may provide recommendations for various disciplines”.

3. Page 3, lines 79-80: Why does this study stand out in relation to other studies on quality of life in elderly people living in low- and middle-income countries? The innovation of the study regarding its contribution to the literature should be clearer. What is the relevance of the study to the context of public health?

Thank you for your suggestions. Previous study identified a major influence of socio-economic status and financial aspect to the quality of life of older adults in LMICs and suggested the needs of further exploration of the relationship between financial aspects and older adults’ QoL, especially in developing countries with rapid aging population . The current study includes both socio-economic status and financial management behaviour as the explanatory variables, which scarcely explored in the previous study. In addition, this study also explored the determinants of physical and mental health aspects of QoL to provide recommendations for various disciplines. This explanation is described in the following lines:

Line 85 – 96: “Quality of life in older adults living in LMICs is associated with socioeconomic, social, psychological and physical factors. However, the relationship of these factors with QoL aspects has scarcely been explored [6,19]. In addition, there is a need for further exploration of the relationship between the financial aspect and QoL in developing countries that have a rapidly ageing population [9]. Therefore, this study aimed to investigate the relationship between the physical and mental aspects of QoL and its determinants among older community-dwelling adults in Indonesia. We hypothesized that individual and environmental characteristics (sociodemographic factors, financial management behaviours, and living arrangements), symptoms (multimorbidity, depression, cognitive impairment, and nutritional status) and functional status (activities of daily living (ADLs) and instrumental activities of daily living (IADLs)) are related to the physical and mental health aspects of QoL in older adults”.

Material and Methods

1. Page 4, lines 94-98: Is it unclear if the study sample representative of the Sleman Regency population of older adults.

Thank you for highlighting this issue regarding our study. The sample of the study were derived randomly from the Sleman HDSS sampling frame. Sleman HDSS’ sampling frame was obtained randomly and systematically from the census blocks of Sleman Regency, thus providing a representative sample from Sleman Regency. However, we understand this will cause selection bias because we did not select the sample directly from the population. We added the detail of Sleman HDSS’ sampling method and the limitation of the study in the following statements:

Line 122 – 126: “The Sleman HDSS collected data from 5147 households selected randomly using two-stage cluster sampling method from the census blocks of Sleman Regency [24]. In 2018, 3129 older adults in the Sleman Regency were registered in the Sleman HDSS sampling frame. The use of the Sleman HDSS sampling frame was aimed facilitating monitoring and follow up for future study”.

Line 377 – 380: “The use of the Sleman HDSS sampling frame and the limited sample in this study also limited our ability to observe association between variables that have been found in other studies. Further studies with larger sample sizes are needed to explore these associations in more detail”.

2. Page 4, lines 98-101: The sampling plan considered the selection of older adults with good quality of life based on a prior study (29). Is this not a source of selection bias? There is a need to better explain this issue, in terms of methods and in terms of discussion.

The use of proportions from previous studies is aimed at identifying the minimum sample to identify the relationship in the population. The proportion was not used to sort or determine the characteristics of the sample in our study. However, we are also aware that this method may limit the sample size in our study, so we may not be able to explore the exact association in the population. We added this description as follows:

Line 377 – 380: “The use of the Sleman HDSS sampling frame and the limited sample in this study also limited our ability to observe association between variables that have been found in other studies. Further studies with larger sample sizes are needed to explore these associations in more detail”.

3. Page 4, lines 104-105: Was this the only exclusion criterion? The criteria for inclusion and exclusion of study participants are not clear.

Thank you for your detailed review. We added the inclusion and exclusion criteria of our study in the following lines:

Line 116 – 119: “The inclusion criteria for this study were older adults 1) aged 50 years and older; 2) living in Sleman Regency; and 3) listed as respondents of the Sleman Health and Demographic Surveillance System (Sleman HDSS), while the exclusion criteria were older adults who declined to participate or were unavailable for two scheduled visits”.

4. Page 5, line 107: What kind of questions were asked to the participants? What is the format of the questions? (semi-structured, open field?)

The questions were close-ended using the standard instruments. The details of the instruments are provided in the exposure variables section when discussing each variable. We added the information of the type of questions in lines 132 – 134: “An approximately 67 minutes face-to-face computer-assisted personal interview (CAPI) was conducted, in which the trained enumerators asked closed-ended questions to older adults and recorded their responses on a digital tablet”.

5. Page 5, lines 118-119: What is the reference of the study that validated the questionnaire for the Indonesian population?

Before the data collection, we conducted a pre-test study to assess the validity and reliability of the study. The description was added in lines 128 – 130: “Before data collection, we conducted a pretest to assess the validity and reliability of the instrument in the study population. The analysis showed a satisfactory result for each instrument, with Cronbach’s α ranging from 0.6–0.8”.

6. Page 6, lines 148-149: What is the reference of the study that validated the questionnaire for the Indonesian population?

Before the data collection, we conducted a pre-test study to assess the validity and reliability of the study. The description was added in lines 128 – 130: “Before data collection, we conducted a pretest to assess the validity and reliability of the instrument in the study population. The analysis showed a satisfactory result for each instrument, with Cronbach’s α ranging from 0.6–0.8”.

7. Page 7, lines 158-159 and lines 175-176: What is the reference of the study that validated the questionnaire for the Indonesian population?

Before the data collection, we conducted a pre-test study to assess the validity and reliability of the study. The description was added in lines 128 – 130: “Before data collection, we conducted a pretest to assess the validity and reliability of the instrument in the study population. The analysis showed a satisfactory result for each instrument, with Cronbach’s α ranging from 0.6–0.8”.

8. Page 7, lines 169-173: What is the definition of multimorbidity adopted by the WHO? I suggest reviewing the concept and including reference at this point in the text.

Thank you for your suggestions. We reviewed and added the definition of multimorbidity by WHO as follows:

Line 197 – 198: “Multimorbidity was defined by the World Health Organization (2016) as “the coexistence of two or more chronic conditions in the same individual” [30]”.

9.Page 8 lines 190-191 and lines 194-195: What is the reference of the study that validated the questionnaire for the Indonesian population?

Before the data collection, we conducted a pre-test study to assess the validity and reliability of the study. The description was added in lines 128 – 130: “Before data collection, we conducted a pretest to assess the validity and reliability of the instrument in the study population. The analysis showed a satisfactory result for each instrument, with Cronbach’s α ranging from 0.6–0.8”.

10. Page 9, lines 209-214: Was the construction of the models done using the stepwise method? It would be interesting to mention the method of adjusting here as well (in the summary says: “We performed stepwise logistic regression analyses to determine the odd ratio (OR) with a 95% confidence interval (CI) ...”).

Yes, our model was constructed using stepwise regression and we edited the manuscript as follows:

Line 233 – 241: “Five models were built using stepwise logistic regression: Model 1 (adjusted for demographic factors); Model 2 (Model 1 adjusted for socioeconomic status and financial management behaviour); Model 3 (Model 2 adjusted for multimorbidity status, nutritional status, and cognitive impairment status); Model 4 (Model 3 adjusted for depression status); and Model 5 (Model 4 adjusted for independence (ADLs and IADLs))”.

Results

1. Page 17, lines 291-300: Could the fact that the majority of the studied sample did not present multimorbidity have influenced this result?

Yes, this condition may be the reason, which the limited sample in our study may cause. We added this reason as the limitation of the study in the following lines:

Line 377 – 379: “The use of the Sleman HDSS sampling frame and the limited sample in this study also limited our ability to observe association between variables that have been found in other studies. Further studies with larger sample sizes are needed to explore these associations in more detail”.

2. Page 17, lines 301-302: The present study is not able to support the conclusion of improvement in physical and mental health through functional independence and absence of chronic diseases. The design cross-sectional does not establish this cause and effect relationship between the outcome and the independent variables (as mentioned in the lines 349-350, Page 19).

Thank you for your detailed reviews. We edited our sentence to avoid the impression of cause-and-effect interpretation as follows:

Line 334 – 338: “Our results indicated that independence in performing ADLs and IADLs and the absence of chronic diseases were correlated with physical health in older adults. In Indonesia, maintaining good physical health may be challenging, as older adult populations experience an upward trend of functional disability in earlier cohorts, which may be caused by the increasing prevalence of multimorbidity [12]”.

Conclusion

1.Would it be interesting to talk more about how this preventive and holistic approach could be carried out in Indonesia? What strategies are already present in primary care that could be strengthened? Are there others that could be implemented?

We provided the discussion of the program that may accommodate these needs in the following line:

Line 353 – 363: “The current program that may accommodate these needs is Integrated Health Care Post for Older Adults. However, this program is mainly focused on monitoring the physical health of older adults and lacks support and participation [54,55]. Evidence from LMICs has shown that bottom-up interventions tailored to the population’s needs and sensitive to the local culture are promising and merit exploration [56], especially in Southeast Asia, where the social, cultural, and economic conditions are so diverse [57]. In Indonesia, because of the collectivist culture, support from community or religious leaders is important to encourage the adoption of a healthy lifestyle and disease management in the community. A case study in Yogyakarta showed that a healthy ageing program could be successful with support from community leaders, shared beliefs of the importance of the program and shared beliefs of nonmaterial rewards of volunteerism [22]”.

Additional Requirements

We edited the style of the manuscript as suggested.

Yes, thank you for the information. We will provide the information on the data accession number when the manuscript is accepted.

---

## [Decision Letter · Decision Letter 1]

10 Oct 2023

PONE-D-23-05720R1Factors related to quality of life in community-dwelling older adults in Indonesia: Results from a cross-sectional studyPLOS ONE

Dear Dr. Kusumaningrum,

Thank you for submitting your manuscript to PLOS ONE. After careful consideration, we feel that it has merit but does not fully meet PLOS ONE’s publication criteria as it currently stands. Therefore, we invite you to submit a revised version of the manuscript that addresses the points raised during the review process.

We look forward to receiving your revised manuscript.

Kind regards,

Fredrick Dermawan Purba, PhD

Academic Editor

PLOS ONE

Reviewers' comments:

Reviewer's Responses to Questions

**Comments to the Author**

1. If the authors have adequately addressed your comments raised in a previous round of review and you feel that this manuscript is now acceptable for publication, you may indicate that here to bypass the “Comments to the Author” section, enter your conflict of interest statement in the “Confidential to Editor” section, and submit your "Accept" recommendation.

Reviewer #3: (No Response)

Reviewer #4: All comments have been addressed

2. Is the manuscript technically sound, and do the data support the conclusions?

Reviewer #3: Partly

Reviewer #4: Yes

3. Has the statistical analysis been performed appropriately and rigorously? 

Reviewer #3: I Don't Know

Reviewer #4: Yes

4. Have the authors made all data underlying the findings in their manuscript fully available?

Reviewer #3: Yes

Reviewer #4: Yes

5. Is the manuscript presented in an intelligible fashion and written in standard English?

Reviewer #3: No

Reviewer #4: Yes

6. Review Comments to the Author

Reviewer #3: I recommend that after this manuscript has been corrected, it be checked again by an English native, especially in the Methods section. In this methods section, some have been written in detail, but others still need improvement, for example, in the instrument section used.

It would be wiser to consider using the word 'Indonesia' in the article title because data collection was only carried out in one region of Indonesia (Sleman, Yogyakarta).

Reviewer #4: The authors have made a great effort to address the editor and the reviewers' comments. Congratulations!

However, I disagree with the statement saying that Indonesia is a country with a rapidly aging population. The percentages of children and labor in Indonesia can be considered as high compared to other countries, such as the Netherlands. The indicative age of invited participants (above 50) can not also be considered as aging. Age above 65 usually is the term used for elderly/aging. I suggest the authors should carefully modify the term used in the article related to aging.

7. PLOS authors have the option to publish the peer review history of their article (what does this mean?). If published, this will include your full peer review and any attached files.

Reviewer #3: No

Reviewer #4: No

---

## [Author Response · Author response to Decision Letter 1]

14 Nov 2023

Comments by Editor

1. Please include a separate legend for each figure in your manuscript.

Thank you. We edited the figure as suggested. 

Comments by Reviewer#3

Title

1. To be honest, when I read the title of this article, I immediately imagined that data collection was carried out in several regions of Indonesia, representing at least six large Indonesian islands or three large regions of Indonesia, for example, West, Central, and East Indonesia. This is my first impression. Shouldn't researchers just mention Yogyakarta? Compared to having to say Indonesia?

Thank you for the reviews and suggestions. We acknowledge the unclear presentation of our study location in the title. Therefore, we change the title as “Factors related to quality of life in community-dwelling adults in Sleman Regency, Special Region of Yogyakarta, Indonesia: Results from a cross-sectional study”.

Abstract

1. My suggestion is that the word 'quality of life' is mentioned only once, and the rest of the time it is abbreviated as ‘QoL’.

We edited the abstract as per suggested.

Introduction

1. “The model is a modification of Wilson and Cleary’s (1995) model that adds a link between individual and environmental characteristics to each concept that influences QoL [18]”. This statement is interesting, but I have not found why it is important to apply this model in this research.

Thank you for your critical suggestion. We clarify it in line 83 – 94.

Line 83 – 94: “In 2005, Ferrans et al. developed a model describing the relationship between QoL and individual and environmental characteristics, biological function, symptoms, functional status, and general health perceptions [20]. Ferrans et al. [20] modified the health-related QoL model by Wilson and Cleary [21] by adding the link between individual and environmental characteristics to each concept that influences QoL, thus providing a more comprehensive description of the relationship between QoL and the associated factors. Given that the Ferrans et al. [20] model provides further understanding of the dynamics of QoL, this study utilized it and hypothesized that individual and environmental characteristics (sociodemographic factors, financial management behaviours, and living arrangements), symptoms (multimorbidity, depression, cognitive impairment, and nutritional status), and functional status (activities of daily living [ADLs] and instrumental activities of daily living [IADLs]) are related to the physical and mental health aspects of QoL”.

Methods

1. Data collection was carried out in 2018, about 5 years ago. Is this data still relevant to the current situation?

Thank you for your detailed reviews. Although our data may not represent the current condition, we thought this study is still relevant and useful based on the following reasons:

• The data related to quality of life in Indonesia are limited. Study which collects data for quality of life and comprehensive variables (IFLS by RAND) provide the most recent data in 2014/2015. In Special Region of Yogyakarta, the routine data collection used the data from primary health centre which may not provide the detailed description of the community. Therefore, considering limited data, this study may provide insight of the condition of QoL in the community-based population, especially in Sleman, Yogyakarta.

• The population of older adults in Indonesia is increasing. Based on the recent data from Central Bureau of Statistics of Indonesia, there are 1.21% increase in older adults’ population between 2018 and 2022. However, Special Region of Yogyakarta is still the province with the greatest share of older adults’ population, which already reach more than 10% in 2018. From the same data, it is also showed that the demographic characteristics of older adults’ population is not changing significantly between 2018 and 2022, in which the majority of older adults’ population are women, in the age of 60 – 69 years old and with low socioeconomic status. From the policy perspective, Special Region of Yogyakarta had just developed an aging policy in 2021, however the implementation is still in preparation. Therefore, considering the characteristics of population, this data may still be relevant to describe older adults. 

• We collected variables which have slow changes in population over the years, such as quality of life and chronic diseases status. Study by Henchoz et al (2019) and Abolhassani et al. (2019) in Switzerland showed that, although there may be some slight decrease in QoL over 5 years, these changes may be more apparent for population with certain characteristics, for example those with disabilities or other health problems. Therefore, in the context of testing the hypothesis, this data may still be relevant to show the association of quality of life and explanatory variables.

• However, as the study was conducted before COVID pandemic, we understand that there may be some changes in the status and dynamics of QoL among older adults’ population, such as found in study by Ahi & Shirzai (2022) or Siltanen et al. (2022). Therefore, we put this as the limitation of the study to encourage the development of further study related the change of quality of life with regard of disaster or other major health problems. 

Considering these reasons, we provide the clarification in line 386 – 388 and 394 – 396.

Line 386 – 388: “Although the data were collected in 2018, the variables collected in this study were considered to have slow changes in the population over the years [65,66]; thus, the data are still relevant for testing our hypothesis”.

Line 394 – 396: “This study was conducted before the COVID-19 pandemic, which may affect the status and dynamics of QoL. Further studies with larger sample sizes are needed to explore these associations in greater detail.”.

2. “In total, 563 older adults aged 50+ years participated in this study”. This sentence is Results and Methods?

Thank you for your detailed reviews. We provide the number 563 in the method section as the results of the calculation of minimum sample size. We revised the sentence to provide better descriptions in line 128 – 132.

Line 128 – 132: “To calculate the minimum sample size in this study, we applied the sample size formula for comparing proportions [27] using the proportions of male (89.4%) and female (80%) older adults with good QoL from a prior study [17], resulting in a total of 488 individuals with 80% power. We added 75 individuals (15% of the total sample) in case of nonresponses and failed interviews, resulting in a minimal sample size of 563 adults”.

3. “The inclusion criteria for this study were older adults 1) aged 50 years and older; 2) living in Sleman Regency; and 3) listed as respondents of the Sleman Health and Demographic Surveillance System (Sleman HDSS), while the exclusion criteria were older adults who declined to participate or were unavailable for two scheduled visits”. From this statement, I would rather recommend that the title mention Sleman, Yogyakarta, so that readers do not respond incorrectly (if Indonesia is mentioned). In my opinion, there would be differences if similar research was carried out on other islands in Indonesia.

Thank you for your suggestion. We changed the title to represent Sleman Regency, Special Region of Yogyakarta, Indonesia.

4. “Before data collection, we conducted a pretest to assess the validity and reliability of the instrument in the study population”. Which instrument is referred to in this statement? My suggestion is that researchers should explain the instrument first before entering this statement so that the reading flow of this manuscript is better.

Thank you for your suggestions. We revised the presentation of our method section to include the subsection of “Instruments” in line 209 – 218.

Line 209 – 218: “We collected the data using a set of standardized instruments consisting of the following: SF-12v2 to assess QoL; the Financial Management Behaviour Scale to measure financial management behaviour; the Mini Nutritional Assessment to assess nutritional status; the Mini Mental State Examination to assess cognitive impairment status; the Geriatric Depression Scale to assess depression status among respondents without cognitive impairment [34]; the Cornell Scale of Depression for Dementia to assess depression status among respondents with severe cognitive impairment [35]; and part of World Health Organization Study on Global Ageing and Adult Health to assess the independence in ADLs and IADLs [36]. Before data collection, we conducted a pretest to assess the validity and reliability of the instrument in the study population. The analysis showed satisfactory results for each instrument, with Cronbach’s α value ranging from 0.6 to 0.8”.

5. I didn't find a subsection about instruments. The research instruments used are explained along with an explanation of 'Outcome variables.' Is it possible to convey this subsection?

Yes. We added the subsection of “Instruments” as explained in the previous answer.

Results

1. A total of 546 participants were included in the analyses. Sampling and Participants:563 older adults aged 50+. Is there an explanation for this difference in participant numbers?

Thank you for your review. We added the explanation in line 250.

Line 250: “Of the 563 respondents at the first visit, 546 were available to participate in the second visit”.

Discussion

1. 'God’s will and eventually accepting their fate (‘pasrah’ or ‘nrimo’)'. One of the things measured in this research is 'mental health'. What do researchers think about aspects of spirituality? I found one study on people with diabetes in Surabaya, Indonesia. Do the researchers think the characteristics of your participants will be the same as the characteristics of the participants in this study? 'Diabetes is a gift from God, a qualitative study coping with diabetes distress by Indonesian outpatients’.

Thank you for your reviews and suggestion. We think that spiritual aspect is important for Indonesian in coping with chronic disease, however, this practice should be complemented with active involvement in disease management to prevent the deterioration of physical health. Yes, the cultural characteristics of the participants between Surabaya and Yogyakarta are almost the same, as both communities are Javanese communities. We edited the description regarding the spiritual aspect and mental health in line 335 – 344.

Line 335 – 344: “When coping with physical health problems, adults in Indonesia rely on religious and spiritual approaches, such as trying to understand the meaning of their conditions, submitting to God’s will and eventually accepting their fate (‘pasrah’ or ‘nrimo’) [51]. These practices were conducted to gain comfort and lessen the effect of illness on their mental states, as shown in patients with diabetes mellitus in Surabaya, Indonesia [52]. Studies have shown that this practice can improve emotional well-being; however, for some individuals, it can also lead to fatalistic attitudes that prevent active involvement in disease management and worsen physical health [51,53]. Therefore, there is a need to promote active involvement in disease management while considering spiritual and religious approaches to maintaining good mental health”.

2. We assessed multimorbidity status based on questions regarding the presence of chronic disease (hypertension, diabetes mellitus, stroke, coronary heart disease, asthma, and chronic obstructive pulmonary disease (COPD)). I think it would be interesting if researchers also added a discussion related to this. I am sure there has been a lot of research in Indonesia regarding this statement. For example, research on HRQoL in people with diabetes, hypertension, etc.

We added the reference from studies in Indonesia about QoL and chronic diseases. We edited the discussion in line 345 – 348.

Line 345 – 348: “Our results indicated that independence in performing ADLs and IADLs and the absence of chronic diseases were correlated with physical health in older adults. This result corroborates the findings of a previous study in Indonesia on QoL among adults with chronic diseases, such as diabetes mellitus [54], chronic obstructive pulmonary disease [55], and cardiovascular diseases [56]”.

Overall

1. I recommend that after this manuscript has been corrected, it be checked again by an English native, especially in the Methods section. In this methods section, some have been written in detail, but others still need improvement, for example, in the instrument section used. It would be wiser to consider using the word 'Indonesia' in the article title because data collection was only carried out in one region of Indonesia (Sleman, Yogyakarta).

Thank you for your suggestion. We edited the Methods section. In addition, the revised manuscript has been reviewed by a professional language editor.

Comments by Reviewer#4

Overall

1. I disagree with the statement saying that Indonesia is a country with a rapidly aging population. The percentages of children and labor in Indonesia can be considered as high compared to other countries, such as the Netherlands. The indicative age of invited participants (above 50) cannot also be considered as aging. Age above 65 usually is the term used for elderly/aging. I suggest the authors should carefully modify the term used in the article related to aging.

Thank you for your suggestion. We refer to the analysis of demographic data in Indonesia by Kudrna, Le & Piggott (2022) which showed the rapid increase in old-age dependency ratio from around 2010s to 2100. However, we revised the wording into “rapid demographic transition” to avoid misconception. We also added the description regarding these claims in line 69 – 73.

Line 69 – 73: “As one of the LMICs with the largest population in Southeast Asia [10], Indonesia is experiencing rapid demographic transition [11]. In 2022, the proportion of older adults was 10.48%, and is projected to reach 19.8% by 2045 [12,13]. In 2100, the old-age dependency ratio will reach approximately 50%, owing to the increase in older adult population and decreasing growth of the younger population [11]”.

Regarding the age of our study participants, we agree that age 50 years cannot be considered as older adults. We included these samples to capture the dynamics of adults who were nearing the retirement age (55 years old in 2018), so that the result can provide recommendations for aging populations in Indonesia, as recommended by Kudrna et al. (2022). However, we revised the terms of “older adults” to “adults” when discussing about the study participants. We added these rationales in line 100 – 102.

Line 100 – 102: “We included a sample of adults aged 50+ years to capture the dynamics of those nearing the retirement age in Indonesia (55 years in 2018) and their influence on QoL, as suggested in a previous study [11]”.

---

## [Decision Letter · Decision Letter 2]

10 Dec 2023

Factors related to quality of life in community-dwelling adults in Sleman Regency, Special Region of Yogyakarta, Indonesia: Results from a cross-sectional study

PONE-D-23-05720R2

Dear Dr. Kusumaningrum,

We’re pleased to inform you that your manuscript has been judged scientifically suitable for publication and will be formally accepted for publication once it meets all outstanding technical requirements.

Kind regards,

Fredrick Dermawan Purba, PhD

Academic Editor

PLOS ONE

Additional Editor Comments (optional):

Reviewers' comments:

Reviewer's Responses to Questions

**Comments to the Author**

1. If the authors have adequately addressed your comments raised in a previous round of review and you feel that this manuscript is now acceptable for publication, you may indicate that here to bypass the “Comments to the Author” section, enter your conflict of interest statement in the “Confidential to Editor” section, and submit your "Accept" recommendation.

Reviewer #3: All comments have been addressed

Reviewer #4: All comments have been addressed

2. Is the manuscript technically sound, and do the data support the conclusions?

Reviewer #3: Yes

Reviewer #4: Yes

3. Has the statistical analysis been performed appropriately and rigorously? 

Reviewer #3: Yes

Reviewer #4: Yes

4. Have the authors made all data underlying the findings in their manuscript fully available?

Reviewer #3: Yes

Reviewer #4: Yes

5. Is the manuscript presented in an intelligible fashion and written in standard English?

Reviewer #3: Yes

Reviewer #4: Yes

6. Review Comments to the Author

Reviewer #3: (No Response)

Reviewer #4: The authors have addressed all the comments made by the reviewer. I don't have any further comments.

7. PLOS authors have the option to publish the peer review history of their article (what does this mean?). If published, this will include your full peer review and any attached files.

Reviewer #3: **Yes: **Bustanul Arifin

Reviewer #4: No

---

## [Editor Report · Acceptance letter]

20 Dec 2023

PONE-D-23-05720R2 

PLOS ONE

Dear Dr. Kusumaningrum, 

I'm pleased to inform you that your manuscript has been deemed suitable for publication in PLOS ONE. Congratulations! Your manuscript is now being handed over to our production team.

Kind regards, 

on behalf of

Dr. Fredrick Dermawan Purba 

Academic Editor

PLOS ONE